# Prognostic value of maximum standard uptake value, metabolic tumor volume, and total lesion glycolysis of positron emission tomography/computed tomography in patients with breast cancer: A systematic review and meta-analysis

**Weibo Wen[1,2], Dongchun Xuan[1], Yulai Hu[2], Xiangdan Li[2], Lan Liu[3]\*, Dongyuan Xu[2]\***

1 Department of Nuclear Medicine, Affiliated hospital of Yanbian University, Yanji, Jilin Province, China,
2 Center of Morphological Experiment, Medical College of Yanbian University, Yanji, Jilin Province, China,
3 Department of Pathology, Affiliated hospital of Yanbian University, Yanji, Jilin Province, China

\* dyxu@ybu.edu.cn (DX); lliu@ybu.edu.cn (LL)

## Abstract

### Purpose

A comprehensive systematic review of the literature was conducted on parameters from [18]F-FDG PET and a meta-analysis of the prognostic value of the maximal standard uptake value (SUVmax), metabolic tumor volume (MTV) and total lesional glycolysis (TLG) in patients with breast cancer (BC).

### Patients and methods

Relevant English articles from PubMed, EMBASE, and the Cochrane Library were retrieved. Pooled hazard ratios (HRs) were used to assess the prognostic value of SUVmax, MTV, and TLG.

### Results

A total of 20 primary studies with 3115 patients with BC were included. The combined HRs (95% confidence interval [CI] of higher SUVmax and higher TLG for event-free survival (EFS) were 1.53 (95% CI, 1.25–1.89, $P = 0.0006$) and 5.94 (95% CI, 2.57–13.71, $P = 0.97$), respectively. Regarding the overall survival (OS), the combined HRs were 1.22 (95%CI, 1.02–1.45, $P = 0.0006$) with higher SUVmax, and 2.91(95% CI, 1.75–4.85, $P = 0.44$) with higher MTV. Higher MTV showed no correlation with EFS [1.31(95% CI, 0.65–2.65, $P = 0.18$)] and similarly higher TLG showed no correlation with OS [1.20(95% CI, 0.65–2.23, $P = 0.45$)]. Subgroup analysis showed that SUVmax, with a median value of 5.55 was considered as a significant risk factor for both EFS and OS in BC patients.

**Data Availability Statement:** All relevant data are within the manuscript and its Supporting Information files.

**Funding:** This research was supported by the National Natural Science Foundation of China (31760330) to D. Xu and the National Natural Science Foundation of China (81560400) and LL.

**Competing interests:** The authors have declared that no competing interests exist.

## Conclusion

Despite clinically heterogeneous BC patients and adoption of various methods between studies, the present meta-analysis results confirmed that patients with high SUVmax are at high risk of adverse events or death in BC patients, high MTV predicted a high risk of death and high TLG predicted a high risk of adverse events.

## Introduction

Breast cancer (BC) is the most common malignancy in women. Although new imaging tools and assisted systemic therapy have improved the survival rate of patients with BC, patients with early invasive BC are still at risk of recurrence or death. It is crucial to identify patients experiencing a risk of relapse or progression, as there is no clinical method for accurate assessment of the prognosis and survival of BC patients till date.

According to the latest report, tumor size, nuclear grade, axillary lymph node involvement, hormone receptor (e.g., estrogen receptor (ER) progesterone) status receptor (PR) and human epidermal growth factor receptor 2 (HER2), and ki-67 proliferation index might act as effective factors in predicting the recurrence or progression in patients at high risk.[1] A growing body of evidence suggests that fluoro18-fluorodeoxyglucose ($^{18}$F- FDG) positron emission tomography (PET/CT) has a great prognostic significance in predicting malignant tumors, TNM staging, evaluation of therapeutic effects, FDG parameter SUV Max, metabolic tumor volume (MTV) and total lesional glycolysis (TLG), as a parameter of tumor metabolism and volume have also received more and more attention. MTV is the size of the tumor tissue, which actively ingests $^{18}$F- FDG, and TLG is the median SUV value in the region of interest MTV [2–5].

However, it is still controversial whether the parameters of $^{18}$F-FDG PET/CT predict the survival rate of BC patients. Some studies reported significant relationships between high SUV max and poor prognoses in patients with BC[6–8], whereas no such correlation is observed by Alexandre Cochet et al. [9]. Therefore, a meta-analysis was designed to evaluate the prognostic value of SUV max MTV and TLG in BC patients.

## Material and methods

### Registration

This systematic review and meta-analysis was reported by following the guidelines of preferred reporting items of the systematic review and meta-analysis (PRISMA) statement[10].

### Inclusion criteria and literature source retrieval strategy

A systematic search of PubMed, Embase, and Cochrane Library (2012-May 2019) using the following keywords ("breast cancer" OR "breast carcinoma")AND("positron emission tomography" OR "positron emission tomography-computed tomography" OR "positron emission tomography computed tomography" OR "PET"OR "PET-CT" OR "PET CT" OR "PET/CT" OR"fluorodeoxyglucose" OR"FDG") AND ("prognostic" OR "prognosis" OR "predictive" OR "survival" OR "outcome") was performed. Inclusion criteria were as follows: (1) studies should include histologically diagnosed BC patients; (2) 18F-FDG PET/CT was used as imaging tool before treatment; (3) the study should at least report one form of survival data; and (4) articles published in English. The exclusion criteria were as follows: (1) studies that focused only on

diagnosis, staging, or monitoring recurrence or progression; (2) studies involving patients with recurrent disease before treatment; and (3) reviews, case reports, conference abstracts and editorial materials. According to the inclusion and exclusion criteria, two authors independently conducted the search and screening, and any discrepancies were resolved by reaching a consensus. If the results reported are from the same sample, completed studies with the latest information will be used.

## Data extraction

Two authors (W Wen and D Xu) independently extracted the following data regarding the included studies (Table 1): (1) basic information of the study, including the year of publication, first author, study time, follow-up duration and study design; (2) details of patients and tumors, including median age, sample size, histology, TNM staging, treatment measures and endpoint. The information regarding 18F-FDG- PET scan data and parameters, determination of fasting time before injection, blood glucose detection before injection, determination of truncated interval value of FDG injection dose, extraction of truncated value of PET parameters SUV Max, MTV, TLG, and tumor profile was also extracted and presented in Table 2.

## Statistical analysis

We followed the same methodology as used in our previous study[28]. Event-free survival (EFS) is defined as the time from treatment initiation to recurrence or progression. In this meta-analysis, disease-free survival (DFS), progression-free survival (PFS), and disease-free metastasis survival in the included studies were combined and redefined as EFS. Overall survival (OS) was defined as the time from therapy initiation till death regardless of the causes[29, 30]. As the effect size of each study, hazard ratio (HR) and 95% confidence interval (CI) take into account the number and time of events, they are considered more accurate and reliable than odds ratio (OR) and relative risk (RR). HR and 95% CI were used to combine the data, and measure the effect of 18F-FDG PET parameters on survival outcome through effect size of HR in order to measure the correlation between SUV max, MTV and TLG values and the prognosis of BC patients. HR is the sum of differences between Kaplan-Meier survival curves and represents comparison of the two groups during a certain follow-up period. The impact of SUV max, MTV and TLG on survival was measured by HR. Data regarding multivariate HR and 95% CI were directly extracted from studies. If multivariate HR was not available, then univariate HR would be extracted. If both multivariate HR and univariate HR were unavailable, then the methodology recommended by Parmar et al[26] would be used to reconstruct HR estimate and its variance based on the survival data from Kaplan-Meier survival curves read by Engauge Digitizer (version 9.4). HR greater than 1 implies worse survival in patients with high SUV max, MTV or TLG, whereas HR less than 1 implies a survival benefit in patients with high SUV max, MTV or TLG.

Statistical heterogeneity was measured using chi-squared Q test and $I^2$ statistic. Heterogeneity was considered to be present if $P<0.05$ or/and $I^2 >50\%$. A fixed effects model was used for meta-analysis when heterogeneity was not significant, while a random effects model was used if heterogeneity was significant. RevMan version 5.3 (RevMan, version 5.3; The Nordic Cochrane Centre, The Cochrane Collaboration) and STATA version 12.0 (STATA Corp., College Station, TX) were used for statistical analysis. Begg's test and Egger's test were used for evaluating bias by STATA version 12.0. P values of less than 0.05 were considered to be statistically significant.

**Table 1. Characteristics of included studies.**

| Study | Year | Country | Study period | Follow-up duration (months) | Median age (range), years | No.of patients | TNM staging | End points | study design | Histology | Treatment |
|---|---|---|---|---|---|---|---|---|---|---|---|
| Selin Carkaci et al. (2013)[11] | 2013 | USA | 2005–2009 | 29(9–55) | 56(33–80) | 53 | I-IV | OS | R | Inflammatory breast cancer | NAC |
| Sung Gwe Ahn et al. (2014)[12] | 2014 | Korea | 2004–2008 | 6.23 year | 48(25–80) | 305 | I-II | RFS | R | NA | CMT/ET |
| Jongtae Cha et al.(2018)[7] | 2018 | Korea | 2008–2013 | 46.2(5.4–95.2) | 50(27–82) | 524 | I-II | RFS | R | IDC/ILC/other | SG+NAC |
| Young Hwan Kim et al. (2015)[13] | 2015 | Korea | 2010–2012 | 28.4(28.4±9.0) | 50.5(30–76) | 119 | II/III | RFS | R | IDC | SG+CMT/ RT/ET |
| Seung Hyup Hyun et al. (2016)[14] | 2016 | Korea | 2006–2012 | 39 | 46.1 ± 10.8 | 332 | I-III | RFS | R | NA | SG/others |
| SUYUN CHEN etal. (2017)[15] | 2017 | China | 2003–2011 | 71(8–118) | 51 | 86 | II/III | EFS OS | R | NA | NAC |
| Ji-hoon Jung et al. (2017)[16] | 2017 | Korea | 2008–2010 | 46 (29–79) | 48.1 (39–79) | 131 | I-III | PFS | R | IDC | SG+CMT/ RT/ET |
| Seung Hyun Son et al. (2014)[9] | 2014 | Korea | 2007–2008 | 53.6(7–66) | NA | 123 | I-III | OS | R | IDC | SG/CMT/Hormone therapy |
| Kazuhiro Kitajima et al. (2017)[6] | 2017 | Japan | 2012–2015 | 32.3 | 57.7 (30–90) | 73 | I-III | DFS | R | IDC/ILC/ others | SG+AC/RT/ET |
| Yannan Zhao et al. (2018)[17] | 2018 | China | 2011–2015 | NA | 59 (37–78) | 27 | IV | PFS | R | NA(MBC) | 500fulvestrant |
| Takayuki Kadoya et al. (2013)[18] | 2013 | Japan | 2006–2011 | 52 | 58.0 ± 12.5 | 344 | I-III | RFS | R | IDC/ILC others | SG |
| Jian Zhang et al.(2013)[19] | 2013 | China | 2007–2010 | 26.6(14.–51.2) | 52(18–70) | 134 | IV | PFS OS | P | NA(MBC) | CMT/RT/Hormone therapy |
| Ana María García Vicente et al.(2015)[20] | 2015 | Spain | 2009–2013 | 34.8 | 54 | 198 | I-III | DFS OS | R | IDC/ ILC/ | SG+NAC |
| Alexandre Cochet et al. (2013)[9] | 2013 | France | 2006–2010 | 30(9–59) | 51(25–85) | 142 | II-IV | PFS | P | IDC/ ILC/ others | NA |
| Suyun Chen rt al.(2015)[21] | 2015 | China | 2006–2011 | 65 (3–106) | 51.9(23–87) | 240 | III-IV | PFS OS | R | IDC/ ILC/ others | SG/CMT/ET |
| Seung Hyun Son et al. (2015)[22] | 2015 | Korea | 2006–2011 | 36.4(0.8–71.4) | 49.1(29–75) | 40 | I-IV | OS | R | IDC | CMT/Hormone therapy |
| Mehdi Taghipour et al. (2015)[23] | 2015 | USA | 2000–2012 | 28.5(0–94) | 60 ± 14 | 78 | I-IV | 0S | R | Ductal carcinoma, Lobular carcinoma Others unknown | SG+CMT |
| Jahae Kim et al.(2012)[24] | 2012 | Korea | 2006–2008 | 50 (17–73) | 52 (32–83) | 53 | I-III | EFS OS | R | IDC/ILC/ others | SG+CMT/ RT/ET |
| Brett Marinelli et al. (2016)[25] | 2016 | USA | 2001–2012 | 12.4 | 54±12 | 47 | NA | OS | R | NA(MBC) | SG+CMT/ RT |
| Jang Yoo, et al.(2017)[26] | 2017 | Korea | 2010–2014 | 30.9(6.6–61.8) | 42.7 (29.5–51.8) | 66 | I-IV | RFS | R | IDC | SG+CMT/ RT |

Abbreviations: NA = not available; R = retrospective; P = prospective; RFS = recurrence/relapse free survival; PFS = progression-free survival; DFS = disease-free survival; OS = overall survival; IDC = invasive ductal carcinoma; ILC = invasive lobular carcinoma; SG = surgery; CMT = chemotherapy; RT = radiotherapy; ET = endocrine therapy; NAC = neoadjuvant chemotherapy; AC = adjuvant chemotherapy.

## Results

### Search results

The search process of the literature was presented in (Fig 1). Search was conducted in three databases, which obtained 559 Embase articles, 1149 PubMed articles and 20 Cochrane Library

**Table 2. Methods of 18 F-FDG PET imaging of included studies.**

| Study | Duration of fasting | Preinjection blood glucose -test | Post-Injection interval | Dose of 18F-FDG | Pet parameters | Determination of cut-off values | Cut-off values | | |
| --- | --- | --- | --- | --- | --- | --- | --- | --- | --- |
| | | | | | | | SUV | MTV(cm 3) | TLG |
| Selin Carkaci et al.(2013) [11] | 6h | <150mg/dL | 70±10 min | (555–629 MBq, 15–17 mCi) | SUV max | Others | 3.8 | | |
| Sung Gwe Ahn et al.(2014) [12] | 8h | <130mg/dL | 60min | 0.14 mCi/kg | SUV max | ROC curve | 4 | | |
| Jongtae Cha et al.(2018)[7] | 6h | <140mg/dL | 60min | 5.5MBq/kg | SUV max | ROC curve | 6.75 | | |
| Young Hwan Kim et al. (2015)[13] | 6h | 150mg/dL | NA | 3–5 MBq/kg | SUV max | ROC curve | 11.1 | | |
| Seung Hyup Hyunet al. (2016)[14] | 6h | 200mg/dL | NA | 5.0 MBq/kg | SUV max | others | 7.0 | | |
| SUYUN CHEN et al.(2017) [15] | 6h | 200mg/dL | 60min | 259–555 MBq | SUV max | ROC curve | 2.5 | | |
| Ji-hoon Jung et al.(2017) [16] | 6h | 150mg/dL | 60min | 4.8MBq/kg | SUV max | ROC curve | 5.5 | | |
| Seung Hyun Son et al. (2014)[27] | 6h | 150mg/dL | 60min | 8.1MBq/kg | SUV max MTV TLG | ROC curve | 5.6 | 8.55 | 14.43 |
| Kazuhiro Kitajima et al. (2017)[6] | 5h | 160mg/dL | 60min | 4.0MBq/kg | SUV max MTV TLG | ROC curve | 3.6 | 3.15 | 16.0 |
| Yannan Zhao et al.(2018) [17] | 6h | 10 mmol/L | 60min | 7.4MBq/kg | SUV max MTV TLG | others | 6.09 | 18.78 | 72.5 |
| Takayuki Kadoya et al. (2013)[18] | 4h | 150mg/dL | 60-90min | 3.7MBq/kg | SUV max | ROC curve | 3.0 | | |
| Jian Zhang et al.(2013)[19] | 6h | 7.8mmol/L | 50-70min | 7.4 MBq/kg | SUV max | ROC curve | 3.54 | | |
| Ana María García Vicente et al.(2015)[20] | 4h | 160 mg/dL | 60min | 370 MBq | SUV max | ROC curve | 6.05 | | |
| Alexandre Cochet et al. (2013)[9] | 6h | NA | 60min | 5MBq/kg | SUV max | others | 5.7 | | |
| Suyun Chen et al.(2015)[21] | 6h | 200mg/dL | 60-90min | 259–555 MBq | SUV max | others | 6.0 | | |
| Seung Hyun Son et al. (2015)[22] | 6h | 150mg/dL | 60min | 8.1MBq/kg | SUV max | ROC curve | 9.4 | | |
| Mehdi Taghipour et al. (2015)[23] | 4h | 200mg/dL | NA | 5.55MBq/kg | SUV max MTV TLG | ROC curve | 2.9 | 7.9 | 11.85 |
| Jahae Kim et al.(2012)[24] | 6h | 8.3mmol/L | 60min | 7.4MBq/kg | SUV max MTV | others | 7.3 | 11.1 | |
| Brett Marinelli et al.(2016) [25] | NA | <200 mg/dL | 60-90min | 444–555 MBq | MTV TLG | others | | NA | NA |
| Jang Yoo, et al.(2017)[26] | 6h | 140mg/dL | 60min | 5.18MBq/kg | TLG | others | | | 52.38 |

Abbreviations: ROC = receiver operating characteristic, SUV max = maximum standard uptake value; MTV = metabolic tumor volume; TLG = total lesional glycolysis; NA = not available.

articles initially (1728 articles). After excluding duplications and meeting summaries, 69 articles that did not meet the inclusion criteria were removed. Finally, 20 studies including 3,115 patients that met the conditions of the study and published from 2013 to 2019 were included in this meta-analysis[6, 7, 9, 11–27] (Fig 1). All 20 studies reported the prognostic values of SUVmax, MTV or TLG for BC survival.

## Literature quality evaluation was included

The quality of 20 studies was assessed according to CRITICAL APPRAISAL OF PROGNOS-TIC STUDIES (https://www.cebm.net/wp-content/uploads/2018/11/Prognosis.pdf), (Fig 2).

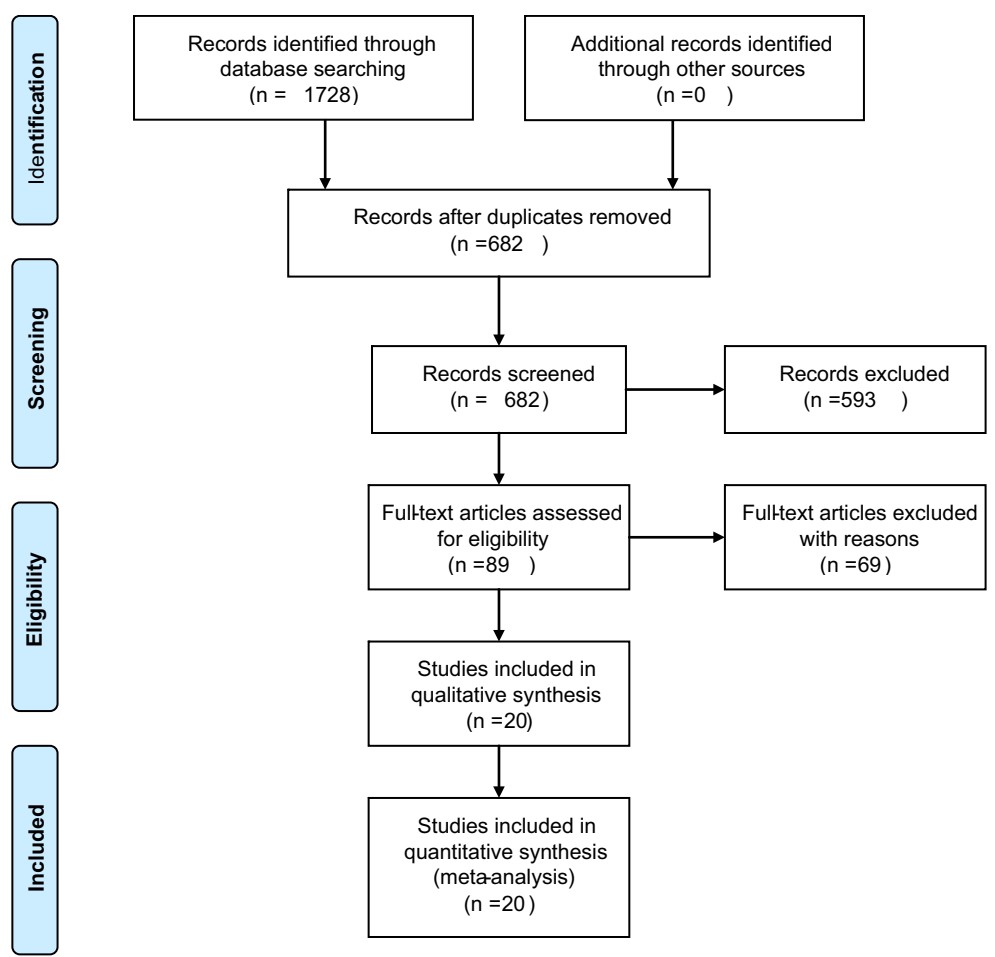

**Fig 1. Flow diagram of study selection.**

Generally, the included studies were of high quality, In the domain of prognostic factor follow-up time measurements, there was a high risk of bias in 7 studies because follow-up data were missing Or the follow-up time was too short. 7 studies were judged to be at high or unclear risk of bias in the domain of defined representative sample measurements because few studies were non-blinded or non-randomized. Most of the studies were well described and monitored regarding adverse events by objective criteria.

## Study characteristics

Almost all the studies were conducted in Asia, with 9 in South Korea, 4 in China and 2 in Japan, 3 in the United States and 1 each in France and Spain. Two were prospective and 18 were retrospective studies. In the 18 SUVmax studies, the SUV cut-off values ranged from 2.5–11.1, which included 14 items with EFS as prognosis and 9 items with OS as prognosis. Among the 6 studies that measured MTV, 3 included EFS as prognosis and 4 included OS as prognosis. Among the 6 studies measuring TLG, 3 with EFS as prognosis and 3 with OS as prognosis were included. In addition, information such as age of the subjects during the follow-up period of tumor pathological staging was extracted. The details of all studies included in the analysis, histology and treatment included in all studies are presented in Table 1. Sixty-

B

A

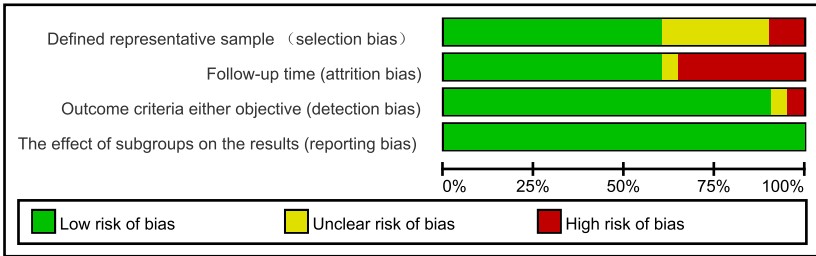

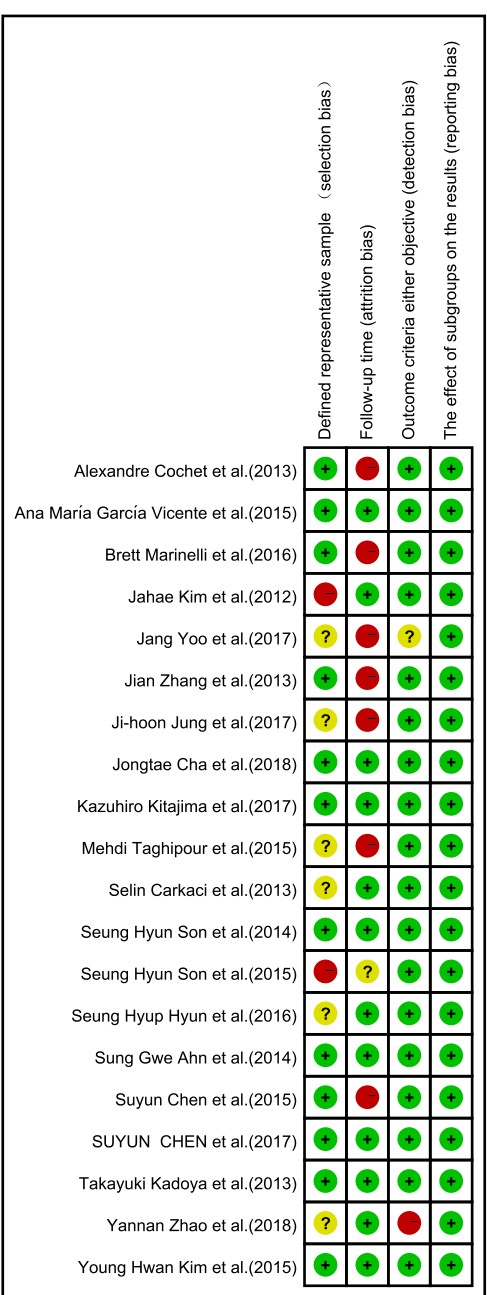

**Fig 2.** (a) Risk of bias graph: review authors' judgments about each risk of bias item presented as percentages across all included studies. (b) Risk of bias summary: judgment of review authors regarding each risk of bias item for each included study.

five percent of patients are with invasive ductal carcinoma (IDC), invasive lobular carcinoma (ILC), and other pathologies. One study [8] involved patients with inflammatory BC, and three studies[12, 17, 27] included only patients with advanced metastatic BC and all these included one or more treatments of surgery (SG)/chemotherapy (CMT)/radiotherapy (RT)/ endocrine therapy (ET)/ hormone /neoadjuvant chemotherapy (NAC). Yannan Zhao et al. 's study[17] involves experimental treatment of metastatic BC with fulvestrant.

## Primary outcome: EFS

Fourteen studies analyzed EFS with SUVmax. After combining HR, the higher SUVmax, and the worse EFS are predicted. Fixed effects model (HR = 1.14; 95% CI = 1.07–1.21, $P$ = 0.0006; $I^2$ = 64%) showed statistical significance, and heterogeneity existed between studies, while random effects model [HR = 1.53; 95% CI = 1.25–1.89, $P$ = 0.0006 (Fig 3A)] still showed meaningful results (Table 3). Potential publication bias was assessed by two statistical tests (Begg's and Egger's). Begg's test showed no significant publication bias ($P$ = 0.352), and Egger's test (S1 Fig) indicated that there might be publication bias ($P$ = 0.002). Therefore, trim and fill analysis was conducted to ensure the reliability of combined HR. Symmetrical funnel plots were obtained after trim and fill analysis (Fig 4). After adding the hypothesis literature, the results were obtained (HR = 1.104; 95% CI: 1.040–1.172), and no substantial change was observed in the results before and after adding the hypothesized literatures, which still showed that SUVmax was significantly correlated with EFS. Sensitivity analysis was conducted to further estimate the impact on combined HRs. Exclusion of each study showed no significant reduction in heterogeneity.

Additional subgroup analyses were performed according to the cutoff method, threshold, analysis method and endpoint (Table 4). Among the studies that included EFS as endpoint, studies that adopted cutoff method using ROC had an HR of 1.57 (95%CI: 1.25–1.97, $P$ = 0.001), and those that adopted cutoff method using other methods showed no statistically significant correlations. According to the median value of SUVmax, the groups of threshold were divided into two subgroups—high ($\geq$5.55) and low (<5.55). Subgroup meta-analyses illustrated that the HRs of SUVmax were 1.20 (95% CI: 1.06–1.35, $P$ = 0.07) and 2.34 (95% CI = 1.22–4.48, $P$ = 0.0004) for high and low cut-off values. For analysis methods, the HRs of studies using univariate analysis was 2.01 (95%CI = 1.36–2.96, $P$ = 0.55), and using multivariate analysis was 1.40 (95%CI = 1.13–1.73, $P$ = 0.004). Based on the endpoint, eligible studies were divided into RFS group, DFS group and PFS group and EFS group, and subgroup analyses results showed that combined HR was 2.02(95% CI: 1.16–3.54, $P$ = 0.005), 2.22(95% CI: 1.02–4.87, $P$ = 0.12), 1.73(95% CI: 1.29–2.32, $P$ = 0.24) and 1.09(95% CI: 1.01–1.17, $P$ = 0.55).

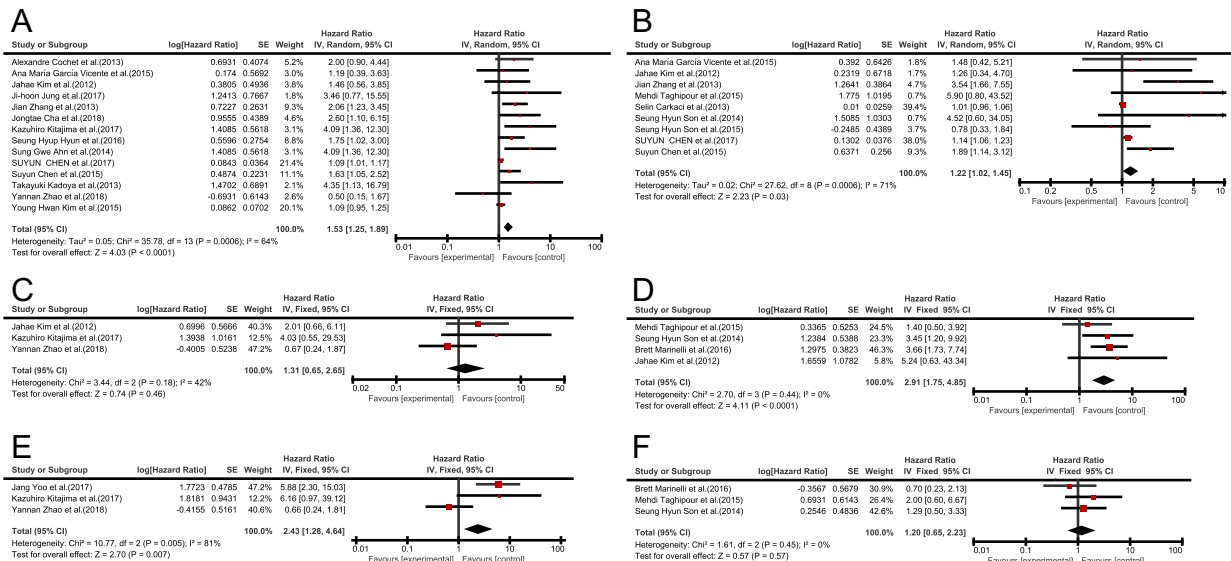

**Fig 3. Forest plots of HR for EFS and OS with SUVmax (A, EFS; B, OS), MTV (C, EFS; D, OS) and TLG (E, EFS;F, OS).** Chi-square test is a measurement of heterogeneity. P < 0.05 indicates significant heterogeneity. Squares = individual study point estimates. Horizontal lines = 95% CIs. Rhombus = summarized estimate and its 95%CI. Fixed: fixed effect model. Random: random effects model.

**Table 3. Summary of meta-analysis results.**

| Endpoint | Metabolic parameter | No.of studies | Model used | HR | 95% CI of HR | P value of HR | Heterogeneity $I^2$ (%) | Conclusion |
|---|---|---|---|---|---|---|---|---|
| EFS | SUV max | 14 | Fixed effect | 1.14 | 1.07–1.21 | 0.0006 | 64 | significant |
| | | | Random effect | 1.53 | 1.25–1.89 | 0.0006 | | significant |
| | MTV | 3 | Fixed effect | 1.31 | 0.65–2.65 | 0.18 | 42 | insignificant |
| | TLG | 3 | Fixed effect | 2.43 | 1.28–4.64 | 0.005 | 81 | significant |
| | | | Random effect | 2.70 | 0.54–13.44 | 0.005 | | insignificant |
| OS | SUV max | 9 | Fixed effect | 1.06 | 1.02–1.10 | 0.0006 | 71 | significant |
| | | | Random effect | 1.22 | 1.02–1.45 | 0.0006 | | significant |
| | MTV | 4 | Fixed effect | 2.91 | 1.75–4.85 | 0.44 | 0 | significant |
| | TLG | 3 | Fixed effect | 1.20 | 0.65–2.23 | 0.45 | 0 | insignificant |

Abbreviations: HR = hazard ratios; CI = confidence interval, EFS = event-free survival ; OS = overall survival; SUV max = maximum standard uptake value;
MTV = metabolic tumor volume; TLG = total lesional glycolysis.

The EFS was based on 3 studies including MTV. A fixed-effects model was used and the pooled HR was 1.31(95% CI 0.65–2.65, $P$ = 0.18; $I^2$ = 42%, Fig 3). There was no significant heterogeneity, and so the results showed no statistically significant correlations.

The EFS was analyzed in 3 studies with TLG. Comprehensive data showed that higher TLG predicted lower EFS, and a fixed effects model (HR = 2.43; 95% CI = 1.28–4.64, $P$ = 0.005; $I^2$ = 81%) showed statistically significant heterogeneity (Fig 3), while random effects model (HR = 2.70; 95% CI = 0.54–13.44, P = 0.005) showed no statistically significant correlations. Sensitivity analysis was carried out to further estimate the impact of combined HRs, and the results revealed that heterogeneity was significantly reduced by the study of Yannan Zhao et al. (2018). [17] ($I^2$ = 0%). The 95% CI of this sensitivity analysis was 2.57–13.71, P = 0.97, which indicated that TLG showed significant correlation with EFS. Potential publication bias was assessed by funnel plots (S2 Fig) and the results revealed no substantial asymmetry.

## Primary outcome: OS

The OS was based on 9 studies including SUVmax. According to the comprehensive data, the higher the SUVmax is, the worse was the OS according to the fixed effects model (HR = 1.06; 95% CI = 1.02–1.10, $P$ = 0.006; $I^2$ = 71%) and random effects model (HR = 1.22; 95% CI = 1.02–1.45, $P$ = 0.006), (Fig 3). Potential publication bias was assessed by two statistical tests (Begg's test and Egger's test). Begg's test ($P$ = 1.000) and Egger's test ($P$ = 0.052), (S3 Fig) showed no significant publication bias. A sensitivity analysis was performed to further estimate the impact of combined HRs and omission of each study showed no significant reduction of heterogeneity.

Additional subgroup analyses were performed according to the cutoff method, threshold and analysis method (Table 4). Among studies that included OS, studies that adopted cutoff method using ROC had an HR of 1.70 (95%CI: 1.07–2.69, $P$ = 0.008), and those that adopted cutoff method using other methods showed no statistically significant correlations. According to the median value of SUVmax, the groups of threshold were divided into two subgroups—high ($\geq$5.55) and low (<5.55). Subgroup meta-analyses illustrated that the HRs of SUVmax had high cut-off values of 1.54 (95% CI: 1.05–2.27, $P$ = 0.38); however, no statistically significant correlations were observed in the HRs with low cut-off values. For analysis methods, the HRs of studies using univariate analysis and multivariate analysis showed no significant results.

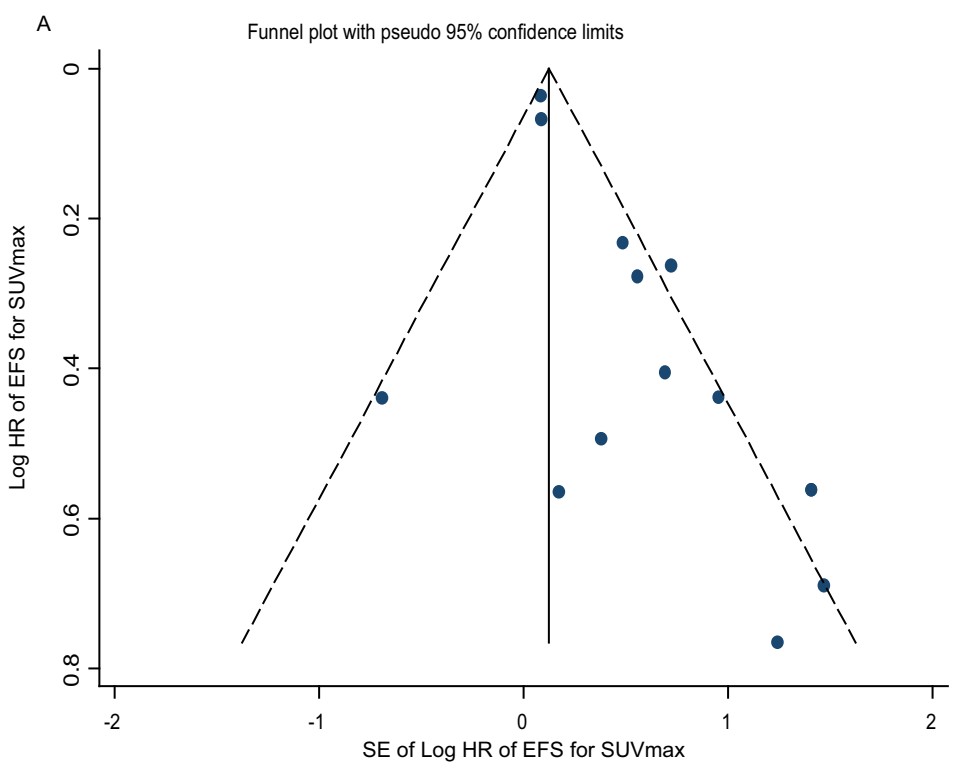

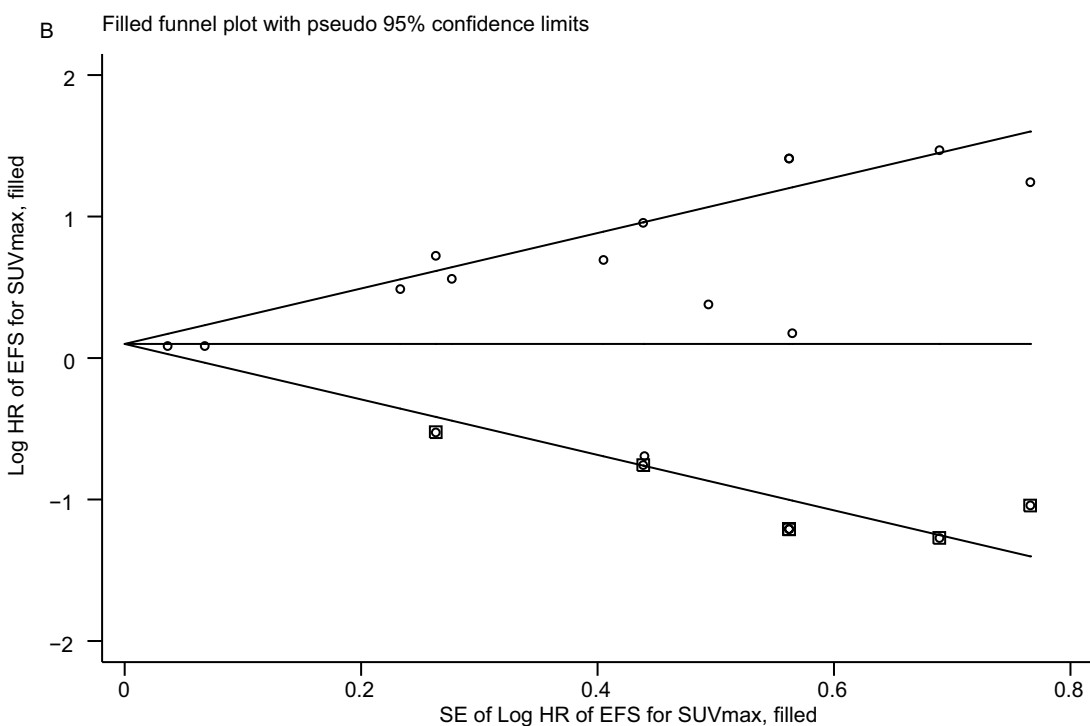

**Fig 4. Funnel plots without (up column) and with (low column) trim and fill.** The pseudo 95% confidence interval (CI) is computed as a part of analysis to produce funnel plots and their corresponding 95%CI for a given standard error (SE). HR indicates hazard ratio.

**Table 4. Subgroup of EFS and OS of SUV max.**

| Endpoint | Volumetric parameters | Factor | No. of studies | Heterogeneity test (I², P) | Effect model | HR | 95%CI of HR | Conclusion |
|---|---|---|---|---|---|---|---|---|
| EFS | SUV max | Cutoff method | | | | | | |
| | | ROC | 11 | 65, 0.001 | random | 1.57 | 1.25,1.97 | significant |
| | | Others | 3 | 42, 0.18 | random | 1.27 | 0.66,2.45 | insignificant |
| | | Threshold | | | | | | |
| | | ≥5.55 | 9 | 44,0.07 | fixed | 1.20 | 1.06,1.35 | significant |
| | | <5.55 | 5 | 80,0.0004 | random | 2.34 | 1.22,4.48 | significant |
| | | Analysis method | | | | | | |
| | | Univariate analysis | 5 | 0,0.55 | fixed | 2.01 | 1.36,2.96 | significant |
| | | Multivariate analysis | 9 | 63,0.004 | random | 1.40 | 1.13,1.73 | significant |
| | | endpoint | | | | | | |
| | | RFS | 5 | 73,0.005 | random | 2.02 | 1.16,3.54 | significant |
| | | DFS | 2 | 58,0.12 | random | 2.21 | 0.66,7.42 | significant |
| | | PFS | 5 | 28,0.24 | fixed | 1.73 | 1.29,2.32 | significant |
| | | EFS | 2 | 0,0.55 | fixed | 1.09 | 1.01,1.17 | significant |
| OS | SUV max | Cutoff method | | | | | | |
| | | ROC | 7 | 66,0.008 | random | 1.70 | 1.07,2.69 | significant |
| | | Others | 2 | 71,0.0006 | random | 1.01 | 0.96,1.06 | insignificant |
| | | Threshold | | | | | | |
| | | ≥5.55 | 5 | 5,0.38 | fixed | 1.54 | 1.05,2.27 | significant |
| | | <5.55 | 4 | 85,0.0002 | random | 1.15 | 0.96,1.38 | insignificant |
| | | Analysis method | | | | | | |
| | | Univariate analysis | 5 | 65,0.02 | random | 1.35 | 0.78,2.35 | insignificant |
| | | Multivariate analysis | 4 | 63,0.04 | random | 1.67 | 0.96,2.89 | insignificant |

Abbreviations: HR = hazard ratios; CI = confidence interval, EFS = event-free survival; OS = overall survival; SUV max = maximum standard uptake value; MTV = metabolic tumor volume; TLG = total lesional glycolysis; RFS = recurrence/relapse free survival; PFS = progression-free survival; DFS = disease-free survival; ROC = receiver operating characteristic.

The OS was analyzed in 4 studies with MTV. Comprehensive data showed that OS prediction remained worse with higher MTV (HR = 2.91; 95% CI = 1.75–4.85, $P$ = 0.44; $I^2$ = 0%, (Fig 3), indicating that MTV was significantly correlated with OS. Potential publication bias was assessed by funnel plots (S4 Fig), and the results showed no substantial asymmetry.

The OS was analyzed in 3 studies with TLG, and fixed effects model was used (HR = 1.20; 95% CI = 0.65–2.23, $P$ = 0.45; $I^2$ = 0%, Fig 3). No significant heterogeneity was observed, and so the results showed no statistically significant correlations.

## Discussion

Clinicians often encounter the problem that many tumors, including breast cancer, are not effectively treated due to lack of standard treatment methods. Doctors and patients then need to reduce the side effects occurred due to failed treatments and avoid unnecessary treatments [31]. PET/CT technology is associated with high sensitivity (96%), strong specificity (94%), and is widely used in diagnosing and staging tumors, with 95% accuracy [32]. SUVmax, MTV and TLG are common parameters of PET/CT used in tumor diagnosis. These metabolic parameters also reflect the biological characteristics of tumors, thus providing some important information regarding clinical prognosis of tumor[33–36]. Patients might benefit if the values of these parameters help in predicting the EFS and OS of BC patients. According to Xia Q

et al. (2015)[37], [18]F- FDG/CT SUVmax value remained very effective in predicting the prognosis of liver metastasis of patients with colorectal cancer. A meta-analysis of 20 published studies was conducted to obtain evidence on the relationship of BC and SUVmax, MTV or TLG. Although SUVmax, MTV and TLG might be affected by varied reasons, our results indicated that patients with high SUVmax are at high risk of EFS along with poorer combined HRs [1.53(95% CI,1.25–1.89, $P$ = 0.0006)] and patients with high TLG is associated with high risk of EFS along with poorer combined HRs [5.94(95% CI, 2.57–13.71, $P$ = 0.97)], while SUVmax and MTV is associated with high risk of OS in patients along with poorer combined HRs [1.22 (95% CI, 1.02–1.45, $P$ = 0.0006)] and [2.91(95% CI, 1.75–4.85, $P$ = 0.44)]. Our meta-analysis results did not reveal the prognostic value of MTV for EFS [HR = 1.31(95% CI 0.65–2.65, $P$ = 0.18), Fig 3] and TLG for OS[HR = 1.20 (95% CI = 0.65–2.23, $P$ = 0.45), Fig 3], as they are influenced by limited sample size, which in turn result in low statistical efficiency. This might be affected by insufficient statistical power, since there were only 3 studies analyzed EFS with MTV and 3 studies analyzed EFS with TLG. Further research should be conducted to investigate the prognostic value of MTV for EFS and TLG for OS in patients with BC.[18] F-FDG-PET/ CT can be used for risk stratification in disease control and survival. Future large-scale prospective studies are warranted to further validate our findings.

Significant heterogeneity was found for SUVmax in predicting EFS and OS. According to the guidelines and protocols for [18] F-FDG PET imaging, heterogeneity of PET/CT parameters (duration of fasting, preinjection blood glucose test, post-injection interval, and dose of [18]F-FDG) included in this study were acceptable as the values were within normal range [3, 38, 39](Table 2). To investigate the source of heterogeneity, cutoff method, threshold, and analysis method were used to conduct subgroup analysis of both EFS and OS, and EFS was used as an endpoint to conduct subgroup analysis (Table 3). Firstly, cutoff method was used to divide the data into two subgroups, in which the ROC group showed no significantly reduced heterogeneity, while others showed no statistically significant correlations. Secondly, the optimal cutoff value of each study was different, and so we divided the study into two groups, with the median value of 5.55. The subgroup with threshold above 5.55 is considered homogeneous ($I^2$ = 44%, $P$ = 0.007). Thirdly, multivariate and univariate methods were adopted to extract HR to study the heterogeneity source of HR. The results showed that heterogeneity between univariate groups was significantly reduced ($I^2$ = 0, $P$ = 0.55). In the four subgroups with different survival endpoints, only two subgroups, PFS ($I^2$ = 28, $P$ = 0.24) and EFS ($I^2$ = 0, $P$ = 0.55), showed significant heterogeneity reduction. So, threshold, source of HR and endpoint are considered to be sources of heterogeneity on EFS. Similarly, cutoff method, threshold, analysis method were used to conduct subgroup analysis on OS. The results showed no reduction in heterogeneity in the subgroups using cutoff method or analysis method. Interestingly, the subgroup with threshold above 5.55 was considered to be homogeneous ($I^2$ = 5%, $P$ = 0.38). Then threshold was regarded as a source of heterogeneity for OS. Our subgroup analysis showed that SUVmax as a significant risk factor for both EFS and OS in BC patients with SUVmax above median value of 5.55, and it is a reason of great satisfaction for us. But we were unable to determine an optimal cut-off value to SUVmax. Different cut-off values and delineation strategies, and various histological methods were used in the studies, which might affect the occurrence of events as well as survival. Further studies with data from individual patients are needed to determine the standard cut-off values and delineation methods for predicting the prognosis using SUVmax.

Significant heterogeneity was also found for TLG in predicting EFS. Three studies confirmed the relationship between TLG and EFS, and fixed effects model showed significant correlation of TLG with EFS. Sensitivity analysis found that the study conducted by Yannan Zhao et al.(2018)[17] was the cause of heterogeneity. Random effects model revealed substantial

changes (HR = 2.70; 95% CI = 0.54–13.44) and showed no statistically significant correlation with EFS. Yannan Zhao et al. (2018) used fulvestrant to treat BC, but it is not a regular BC treatment drug. No similar study was found in our research. Several larger sample size studies are needed in future to find out whether fulvestrant has an impact on EFS in BC patients with high TLG. After discussion, we considered that this study would not be included as prognosis of EFS in BC patients has high TLG for time being.

Moreover, the quality of included studies should also be taken into account as it is a limitation of our study. Firstly, although all the included studies were evaluated by Cochrane risk bias tool and included high quality studies, some studies still lacked partial details of patient and data of $^{18}$F-FDG PET scan. Furthermore, prospective studies combining survival rate of BC and PET parameters are needed. Secondly, BC is a heterogeneous disease, and patients with different histological grades, stages, and treatments were included in this meta-analysis, which can affect the events occurring over time and survival. Thirdly, as far as we know, there are some studies on PET parameters of tumors or lymph nodes, but our study focused only on tumor parameters. Fourthly, non-English articles were excluded in this study, which might lead to potential impact of language bias. Fifthly, only published studies were included when searching the electronic databases, and so publication bias cannot be excluded. However, evaluation of publication bias suggested that our analysis was reliable. Sixthly, Engauge Digitizer was used to extract the data of HRs from survival curves indirectly, leading to an imprecision. Finally, studies included in this meta-analysis are almost conducted in Asia, and the incident of BC is high in these regions and race of humans in these countries might cause bias.

## Conclusion

Despite the adoption of different methods for different types of BC patients, the present meta-analysis confirmed that BC patients with high SUVmax are at high risk of adverse events or even death, while MTV is associated with high risk of death and TLG is associated with high risk of adverse events. However, our meta-analysis did not reveal the prognostic value of MTV for adverse events and TLG for death.

## Supporting information

**S1 Fig. Egger's test of SUV max and EFS.**
(EPS)

**S2 Fig. Funnel plots of EFS and TLG.**
(EPS)

**S3 Fig. Egger's test of SUV max and OS.**
(EPS)

**S4 Fig. Funnel plots of MTV and OS.**
(EPS)

**S1 File. PRISMA checklist.**
(DOC)

## Author Contributions

**Conceptualization:** Weibo Wen, Dongyuan Xu.

**Data curation:** Weibo Wen, Lan Liu, Dongyuan Xu.

**Formal analysis:** Weibo Wen.

**Funding acquisition:** Lan Liu, Dongyuan Xu.

**Investigation:** Weibo Wen, Dongyuan Xu.

**Methodology:** Xiangdan Li, Lan Liu, Dongyuan Xu.

**Project administration:** Weibo Wen, Xiangdan Li.

**Resources:** Xiangdan Li, Lan Liu, Dongyuan Xu.

**Software:** Weibo Wen, Dongchun Xuan.

**Supervision:** Lan Liu, Dongyuan Xu.

**Validation:** Dongchun Xuan, Yulai Hu.

**Visualization:** Yulai Hu, Dongyuan Xu.

**Writing – original draft:** Dongyuan Xu.

**Writing – review & editing:** Weibo Wen, Dongyuan Xu.

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
