## [Decision Letter · Decision Letter 0]

1 Nov 2019

PONE-D-19-26087

Prognostic value of maximum standard uptake value, metabolic tumor volume, and total lesion glycolysis of positron emission tomography/computed tomography in patients with breast cancer: A systematic review and meta-analysis

PLOS ONE

Dear Dr Xu,

Thank you for submitting your manuscript to PLOS ONE. After careful consideration, we feel that it has merit but does not fully meet PLOS ONE’s publication criteria as it currently stands. Therefore, we invite you to submit a revised version of the manuscript that addresses the points raised during the review process.

We would appreciate receiving your revised manuscript by Dec 16 2019 11:59PM. To enhance the reproducibility of your results, we recommend that if applicable you deposit your laboratory protocols in protocols.io, where a protocol can be assigned its own identifier (DOI) such that it can be cited independently in the future. For instructions see: http://journals.plos.org/plosone/s/submission-guidelines#loc-laboratory-protocols

We look forward to receiving your revised manuscript.

Kind regards,

Giorgio Treglia, MD, MSc

Academic Editor

PLOS ONE

Journal Requirements:

This research was supported by the National Natural Science Foundation of China (31760330).

Please remove any funding-related text from the manuscript and let us know how you would like to update your Funding Statement.

3. Please complete your Competing Interests on the online submission form to state any Competing Interests. If you have no competing interests, please state "The authors have declared that no competing interests exist.", as detailed online in our guide for authors at http://journals.plos.org/plosone/s/submit-now This information should be included in your cover letter; we will change the online submission form on your behalf.

4. Please ensure that you include a title page within your main document. You should list all authors and all affiliations as per our author instructions and clearly indicate the corresponding author.

Reviewers' comments:

Reviewer's Responses to Questions

**Comments to the Author**

1. Is the manuscript technically sound, and do the data support the conclusions?

Reviewer #1: Yes

Reviewer #2: Yes

2. Has the statistical analysis been performed appropriately and rigorously? 

Reviewer #1: Yes

Reviewer #2: Yes

3. Have the authors made all data underlying the findings in their manuscript fully available?

Reviewer #1: Yes

Reviewer #2: Yes

4. Is the manuscript presented in an intelligible fashion and written in standard English?

Reviewer #1: Yes

Reviewer #2: Yes

5. Review Comments to the Author

Reviewer #1: Authors investigated the prognostic value of SUVmax, MTV and TLG in patients with breast cancer through a systematic review and meta-analysis of previous studies.

The topic is interesting and the work is methodologically good, nevertheless few major concerns emerged.

1. I understand that MTV and TLG have not a significant correlation with PFS and OS in the meta-analysis, but authors did not clearly explain this data in the results and discussion section

2. Only SUVmax significantly correlate with PFS and OS, so authors should rewrite the paper according to this main significant result.

Reviewer #2: Dear editor,

Thank you for giving me the opportunity to review for your journal.

The article under review is a very well performed systematic review and meta-analyses on prognostic significance of FDG PET/CT parameters in breast cancer patients.

Systematic reviews of prognostic studies are very hard to perform and the authors should be commended in this regard.

The search strategy is OK, inclusion and exclusion criteria are appropriate.

The statistical analyses are appropriate.

The quality assessment is not appropriate. Actually the authors used the quality assessment tools for treatment studies. For prognostic studies, there are couple of quality assessment tools for example the following link can be used

https://www.cebm.net/wp-content/uploads/2018/11/Prognosis.pdf

In my opinion the study can be published with minor revision.

Best wishes,

Ramin Sadeghi ,MD

6. PLOS authors have the option to publish the peer review history of their article (what does this mean?). If published, this will include your full peer review and any attached files.

Reviewer #1: No

Reviewer #2: Yes: Ramin Sadeghi, MD

---

## [Author Response · Author response to Decision Letter 0]

7 Nov 2019

Response to the Reviewers

Point by point reply to the comments

Answer to Reviewer #1:

1. I understand that MTV and TLG have not a significant correlation with PFS and OS in the meta-analysis, but authors did not clearly explain this data in the results and discussion section

Answer: Thank you for your valuable advice. I have supplemented the data in the results section and the discussion section according to your requirements, It is also marked in red in the original text.

In the results section: The EFS was based on 3 studies including MTV. A fixed-effects model was used and the pooled HR was 1.31(95% CI 0.65-2.65, P=0.18; I2 = 42%, figure 3B). There was no significant heterogeneity, and so the results showed no statistically significant correlations. The OS was analyzed in 3 studies with TLG, and fixed effects model was used (HR =1.20; 95% CI = 0.65-2.23, P=0.45; I2 = 0%, figure 3F). No significant heterogeneity was observed, and so the results showed no statistically significant correlations.

 In the discussion section: Our meta-analysis results did not reveal the prognostic value of MTV for EFS [HR=1.31(95% CI 0.65-2.65, P=0.18) , figure 3B] and TLG for OS[HR=1.20 (95% CI = 0.65-2.23, P=0.45), figure 3F].

2. Only SUVmax significantly correlate with PFS and OS, so authors should rewrite the paper according to this main significant result.

Answer: Thank you for your valuable advice. I have rewritten the discussion of MTV for EFS and TLG for OS according to your requirements. Additional discussion sections are as follows: Our meta-analysis results did not reveal the prognostic value of MTV for EFS [HR=1.31(95% CI 0.65-2.65, P=0.18) , figure 3B] and TLG for OS[HR=1.20 (95% CI = 0.65-2.23, P=0.45), figure 3F], as they are influenced by limited sample size, which in turn result in low statistical efficiency. This might be affected by insufficient statistical power, since there were only 3 studies analyzed EFS with MTV and 3 studies analyzed EFS with TLG. Further research should be conducted to investigate the prognostic value of MTV for EFS and TLG for OS in patients with BC.18 F-FDG-PET/CT can be used for risk stratification in disease control and survival. Future large-scale prospective studies are warranted to further validate our findings.

Answer to Reviewer #2:

Thanks for your valuable comments on my paper. I have benefited a lot. The following is the answer to your question: The quality assessment is not appropriate. Actually the authors used the quality assessment tools for treatment studies. For prognostic studies, there are couple of quality assessment tools for example the following link can be used

https://www.cebm.net/wp-content/uploads/2018/11/Prognosis.pdf

Answer: The following is what I modified and added according to your requirements and it is also marked in red in the original text. We also modified the quality evaluation chart. The rewrite is as follows: The quality of 20 studies was assessed according to CRITICAL APPRAISAL OF PROGNOSTICSTUDIES

(https://www.cebm.net/wp-content/uploads/2018/11/Prognosis.pdf)(figure2). Generally, the included studies were of high quality，In the domain of prognostic factor follow-up time measurements , there was a high risk of bias in 7 studies because follow-up data were missing Or the follow-up time was too short. 7 studies were judged to be at high or unclear risk of bias in the domain of defined representative sample measurements because few studies were non-blinded or non-randomized. Most of the studies were well described and monitored regarding adverse events by objective criteria.

---

## [Decision Letter · Decision Letter 1]

18 Nov 2019

Prognostic value of maximum standard uptake value, metabolic tumor volume and total lesion glycolysis of positron emission tomography/computed tomography in patients with breast cancer: A systematic review and meta-analysis

PONE-D-19-26087R1

Dear Dr. Xu,

We are pleased to inform you that your manuscript has been judged scientifically suitable for publication and will be formally accepted for publication once it complies with all outstanding technical requirements.

With kind regards,

Giorgio Treglia, MD, MSc

Academic Editor

PLOS ONE

Additional Editor Comments (optional):

Reviewers' comments:

Reviewer's Responses to Questions

**Comments to the Author**

1. If the authors have adequately addressed your comments raised in a previous round of review and you feel that this manuscript is now acceptable for publication, you may indicate that here to bypass the “Comments to the Author” section, enter your conflict of interest statement in the “Confidential to Editor” section, and submit your "Accept" recommendation.

Reviewer #1: All comments have been addressed

Reviewer #2: All comments have been addressed

2. Is the manuscript technically sound, and do the data support the conclusions?

Reviewer #1: Yes

Reviewer #2: Yes

3. Has the statistical analysis been performed appropriately and rigorously? 

Reviewer #1: Yes

Reviewer #2: Yes

4. Have the authors made all data underlying the findings in their manuscript fully available?

Reviewer #1: Yes

Reviewer #2: Yes

5. Is the manuscript presented in an intelligible fashion and written in standard English?

Reviewer #1: Yes

Reviewer #2: Yes

6. Review Comments to the Author

Reviewer #1: Authors changed the paper according to Reviewers' comments.

The manuscript clearly improved and could be now accepted.

Reviewer #2: (No Response)

7. PLOS authors have the option to publish the peer review history of their article (what does this mean?). If published, this will include your full peer review and any attached files.

Reviewer #1: No

Reviewer #2: Yes: Ramin Sadeghi, MD Associate Professor of Nuclear Medicine, Mashhad University of Medical Sciences, Mashhad, Iran

---

## [Editor Report · Acceptance letter]

26 Nov 2019

PONE-D-19-26087R1 

Prognostic value of maximum standard uptake value, metabolic tumor volume, and total lesion glycolysis of positron emission tomography/computed tomography in patients with breast cancer: A systematic review and meta-analysis 

Dear Dr. Xu:

I am pleased to inform you that your manuscript has been deemed suitable for publication in PLOS ONE. Congratulations! Your manuscript is now with our production department. 

With kind regards,

on behalf of

Dr. Giorgio Treglia 

Academic Editor

PLOS ONE